# Determinants of advanced age pregnancy in Ethiopian; multi-level analysis of Ethiopian demographic health survey 2016

**Aynamaw Embiale Tesega**[1]*, **Aynadis Enyew**[2], **Degefa Gomora Tesfaye**[1], **Girma Geta**[1], **Muche Argaw**[3], **Alamirew Enyew Belay**[4]

**1** Midwifery Department, College of Medicine and Health Science, Madda Walabu University, Bale Goba, Ethiopia, **2** Marie Stopes International Ethiopia Shashemene BEmONC Center, Shashemene, Ethiopia, **3** Midwifery Department, College of Medicine and Health Science, Wolkite University, Wolkite, Ethiopia, **4** Surgical Nursing Department, College of Medicine and Health Science, Bahir Dar University, Bahir Dar, Ethiopia

* aembiale@gmail.com

## Abstract

### Background

Advanced maternal-age pregnancy has become a serious public health problem in both developed and developing countries due to adverse birth outcomes for the mother, fetus, or newborn. However, there are limited studies conducted to identify determinants of advanced-age pregnancy in Ethiopia. Therefore, this study aimed to assess individual and community-level determinants of advanced age pregnancy in Ethiopia.

### Methods

This study was based on 2016 Ethiopian Demographic and Health Survey data. Three thousand two hundred ninety-two weighted samples of pregnant women were included in this analysis. A multilevel logistic regression model was conducted to assess the determinants of advanced-age pregnancy among the study participants in Ethiopia.

### Results

maternal age at first birth (AOR = 4.05, 95% CI: 1.77–9.22), level of maternal education [primary education 2.72 times (AOR = 2.27, 95 CI: 1.55–4.76) and secondary and above education (AOR = 5.65, 95% CI: 1.77–17.70)], having a history of alcohol (AOR = 11.8, 95% CI: 5.71–24.42), parity (AOR = 3.22, 95% CI: 2.69–3.84), number of household member (AOR = 1.22, 95% CI: 1.05–1.41), family planning unmet need for spacing of pregnancy (AOR = 4.79, 95% CI: 2.63–8.74), having sons/daughters elsewhere (AOR = 1.89, 95% CI: 1.22–2.94), had higher community poverty level (AOR = 2.37, 95% CI: 1.16–4.85), those had higher community unmet need for family planning (AOR = 5.19, 95% CI: 2.72–9.92) were more likely to have advanced age pregnancy. Whereas Living in an Emerging region (AOR = 0.29, 95% CI: 0.14–0.59) and living in a metropolitan city (AOR = 0.03, 95% CI: 0.03–0.38), were less likely to have advanced age pregnancy.

**Data Availability Statement:** data we used for this study was attached as supplementary file titled as "our last data for the analysis" in STATA form.

**Funding:** The author(s) received no specific funding for this work.

**Competing interests:** The authors have declared that no competing interests exist

## Conclusions

increased Maternal age at first birth, level of maternal education, history of alcohol drinking, increased number of parity and household members, family planning unmet need for spacing, had sons/daughters elsewhere, had higher community poverty level, those had higher community unmet need for family planning positively, whereas living in the emerging region and living in metropolitan's city was negatively affect advanced age pregnancy. Help women to have informed decision-making and create platforms to women have special care during this age of pregnancy. Empower women on family planning and socioeconomic status.

## Background

Advanced maternal age is usually defined as being 35 years or older which is believed to predispose mothers to enormous adverse outcomes during pregnancy and delivery [1]. According to Alford and his colleagues, It is defined as the mother's age of 35 years or over at delivery, associated with an increased risk of Severe Maternal Mortality [2]. Center for Disease Control and Prevention (CDC) data by 2020 demonstrates the continued upward trend in the mean age of pregnancy was 19% and 11% from all pregnancies and first pregnancies among women aged 35 years and older [3].

A population-based retrospective cohort study in America from 1998–2013 revealed out of 690,471 total births 105,067(15.2%) were in women 35 years or greater [4]. In a retrospective observational case-control study conducted in Turkey, from a total of 1202 delivered mothers who had given birth from June 2016 to December 2017, 632 were advanced maternal age pregnancies [5]. Similar studies in Oman and Ireland Advanced Age Pregnancy were 22.4% and 18.18% respectively [6, 7]. In a facility-based prospective cohort study conducted in Ghana Teaching Hospital from 175 normal pregnant women 58 (33.14%) were 35–45 years old [8]. In another Facility-based cross-sectional Multi-country Survey; the prevalence of pregnant women with Advanced Age was 12.3% from pregnant women 308,149 [9].

Advanced maternal age is associated with pregnancy-induced hypertension, pre-pregnancy, and gestational diabetes related to decreasing pancreatic beta cell function, increasing body mass index, and decreasing insulin sensitivity and Antepartum hemorrhage secondary to Placenta previa and abruption commonly occurred due to higher parity, age-related endothelial damage, and prior uterine surgery, maternal near-miss, increased cesarean delivery, mal-presentation, and maternal death. Alongside this, Advanced Maternal Age also results in neonatal complications, such as Low Appearance, Pulse, Grimace, Activity, and Respiratory score, higher Neonatal Intensive Care Unit admission, preterm delivery, low birth weight, birth defects, chromosomal abnormalities, stillbirth, and perinatal death [10–26].

Studies done in Ethiopia show that advanced maternal age was found to be significantly associated with higher odds of adverse pregnancy outcomes; those adverse maternal outcomes indicated, a higher rate of cesarean delivery, Antepartum Hemorrhage, pregnancy-induced hypertension, Diabetes Miletus, as witnessed by studies from Jimma University specialized Hospital, Arba Minch zuria, and Gacho Baba district, and Debre Tabor town [15, 16, 27–32]. Whereas adverse perinatal outcomes were a higher risk of delivering LBW newborns compared to a maternal age of 30–34 [33].

A community-based prospective cohort study conducted in Arba Minch showed stillbirth and neonatal mortality were significantly associated with advanced maternal age [30]. A

comparative cross-sectional study in northern Ethiopia revealed that advanced maternal-age pregnancy was significantly associated with adverse perinatal outcomes like preterm delivery, low birth weight, perinatal death, and low fifth-minute Appearance, Pulse, Grimace, Activity, and Respiratory score [31].

Many societal factors have facilitated this trend including, late marriage, higher education, effective birth control, and advances in assisted reproductive technology. With the progress of assisted reproductive technology; it is now possible to extend a woman's reproductive life beyond the age of natural menopause [34]. It is supported by another study done in the United States of America as demographic data show an increased population of women aged 35–45 influenced by evolving social and cultural changes, including higher rates of divorce, having multiple partners before settling down, living together before marriage, and having a later or second marriage. Women with higher socioeconomic status (SES) and higher levels of education tend to delay motherhood into their mid-to-late 30 [35].

The international academic community has extensively explored the association of Advanced Maternal Age with adverse obstetrical and perinatal outcomes [36, 37]. However, up to the investigator's knowledge, no research has been conducted to assess advanced age pregnancy determinants in Ethiopia as a country. Since this group was significantly associated with a higher risk of unwanted pregnancy [38]. Hence, there is a need to research the determinants of advanced-age pregnancy from Ethiopian DHS 2016.

## Methods and materials

### Study design, period, and setting

Ethiopian DHS 2016 was a cross-sectional collected data set, which was used in this study. The data was collected from January 18 to June 27, 2016. Ethiopia is located in the horn of Africa and is the second most populous country in the continent, with over 100 million inhabitants (2017 estimates). The country is currently divided into 13 administrative divisions—11 regional states and 2 city administrations (Addis Ababa which is the capital city of the country, & Dire Dawa), but this Ethiopian DHS was done in nine regions and two city administrations. The study target groups were pregnant women in randomly selected households across the country.

### Source of data

Secondary data was used for this study from the 2016 Ethiopia Demographic and Health Survey (2016 EDHS). The survey collected demographic and health information from a nationally representative sample of women in the reproductive age group 15–49 and men aged 15–59. It was a periodic cross-sectional survey conducted by the Central Statistics Agency (CSA), which was funded by the United States Agency for International Development (USAID) in many middle- and low-income countries. It is available in the DHS database http://dhsprogram.com/data/available-datasets.com. DHS granted permission to access data through the project title "Determinants of Advanced Age Pregnancy in Ethiopian DHS 2016."

### Sampling procedure

The sample was stratified and selected in two stages. In the first stage, a total of 645 EAs (202 EAs in urban areas and 443 EAs in rural areas) were selected with probability proportional to the EA size and with independent selection in each sampling stratum. Stratification is the process by which the sampling frame is divided into subgroups or strata that are as homogeneous as possible using certain criteria like regions, place of residence (rural and urban). In the

second stage of selection, a fixed number of 28 households per cluster were selected with an equal probability of systematic selection from the newly created household listing. The individual response that was fully collected was eligible for the study. On the contrary, those whose individual responses were not full were excluded from the study and counted as missing data in the analysis process.

## Weighting data

Sampling weights are adjustment factors applied to each case in tabulations to normalize the differences in probability selections and interviews between cases in the sample due to its design. The EDHS samples were not self-weighted due to the need for data for specific regions or areas of the country that often need to be over-sampled. Thus, we applied sample weights to the data when we computed the analyses.

## Study population

All women aged 15–49 and all men aged 15–59, who were either permanent residents of the selected households or visitors who stayed in the household the night before the survey, were eligible to be interviewed. Current pregnancy is one of the sections that have been covered in each subsequent year in the nine regions and two administrative regions both at rural and urban levels. The EDHS data of 2016 included a weighted sample of 3,292 pregnant women in the data collection period, which we used in this study.

## Dependent variable

In this study the dependent variable was advanced age pregnancy in 2016 Ethiopian DHS; it is coded as "1" if the pregnant woman was aged 35 and above years old, while coded as "0" if the pregnant woman was less than 35 years old. Therefore, the $i^{th}$ pregnant woman is represented by a random variable $Yi$, with two possible values coded as "1" and "0".

## Independent variables

Independent variables that were used as predictors of advanced-age pregnancy included in this study were selected based on the two-level weighted binary logistic regression model (multi-level multivariate logistic regression). These variables were categorized as individual and community-level factors.

## Individual level variables

These were variables to the individual woman, her partner, or their household related. These were; the mother's religion(Muslim, Orthodox, and other religions (protestant, catholic, and other traditional), maternal and partner educational level (no education, have formal education), maternal and partner occupations (agrarian, employed, and workless), wealth index (poor, middle, and rich), sex of household head (male, female), relationship to household head (husband, wife, and others), the decision maker at the house hold level (woman/her partner), list of household members, number of under-five at home, sons and daughters (at home, elsewhere and died), birth order (parity), history of alcohol drinking (yes/no) and chat chewing (yes/no), media exposure (television, radio magazines, internet), history of abortion (yes, no), family planning unmet need (no unmet need, unmet need for spacing and unmet need for limiting), and age at first (sex, marriage, and delivery).

## Community level variables

1. **Contextual regions;** Agrarian (Tigray, Amhara, Oromia, SNNP), emerging region (Afar, Somali, Benishangul-Gumuz, and Gambela), and metropolitan cities (Addis Ababa, Dire Dawa, and Harari) [39].

2. **Residence (urban, rural);** was taken as it is from the Ethiopian demographic health survey data without any modifications [40].

3. **Community illiteracy:** This is the aggregate value of the educational illiteracy of women based on the average proportions of educational levels in the community. The aggregate could show the overall pregnant women educational illiteracy of the clusters. There were two values for this variable, concerning the median value of 0.5; those having a median value ≥0.5 were labeled as higher community educational illiteracy, and less than <0.5 were labeled as lower community educational illiteracy [41–43].

4. **Community poverty;** this variable was created from median values of wealth index categories of the individual mothers. The two values for the community poverty level were higher community poverty if the median value≥0.5 and lower community poverty if the median value <0.5 [41].

5. **Community-women empowerment;** the aggregate values of the variable of women's empowerment were computed based on median values of the aggregates and were coded as higher community empowerment if the median value≥0.5 and lower community empowerment if the median value <0.5 [42].

6. **Community Media unexposed;** with a similar approach, this variable was derived from the individual responses for exposure to radio or television since most of the pregnant women were illiterate. We defined the community media as higher unexposed if the median value≥0.5 and lower media unexposed if the median value <0.5 [41, 43].

7. **Community family planning unmet need;** this was the aggregated value of individual-level variable family planning unmet need (unmet need from limiting and spacing). This is also categorized as higher community unmet need if the median value≥0.5 and lower community unmet need if the median value <0.5 [43, 44].

## Statistical analysis

Data were checked for missing values and data cleaning, exploratory analysis, variable recoding, labeling, categorization, and re-categorization were performed prior to analysis. The analysis was done using Stata version 14 and the presence of collinearity among independent variables was checked through variance inflation factor (VIF) taking a cut-off value of 10. However, the VIF value for all predictors was less than 10, indicating that there was no multicollinearity between variables. Descriptive analysis was carried out using frequencies and percentage distributions of the sample for each of the variables. We use the birth record (BR) file with a total of weighted 3,292 sampled pregnant women for data analysis.

To account for the underestimation of the standard errors of the final estimates, we analyzed the crude associations of individual and community factors with advanced-age pregnancy by considering the sample design of the Ethiopian demographic health survey using the cluster variable. Both bi-variable and multi-variable multilevel logistic regression were performed to assess the independent effect of the individual and community level variables on advanced maternal age pregnancy. Independent variables with p-values less than 0.25 in the

bi-variable multi-level analysis were entered into the multivariable multi-level logistic regression analysis.

## Multilevel regression analysis

Since demographic health survey data were hierarchical, i.e. mothers are nested within households, and households are nested within clusters, the use of flat models could underestimate standard errors of the effect sizes. With such data, mothers within a cluster may be more similar to each other than mothers in the rest of the country. This violates the assumption of the flat model's independence of observations and equal variance across the clusters. This implies a need to consider the between-cluster variability. All these issues motivated us to use multilevel modeling, which was able to compute a fixed effect for both the individual and community factors and a random effect for the between-cluster variation simultaneously. The fixed-effect sizes of individual and community-level factors on advanced-age pregnancy were expressed using the Odds Ratio (OR) with 95% confidence intervals (95% CI) and p-value less than 0.05.

As the responses to pregnant women as advanced and non-advanced were dichotomous, we opted to run a two-level mixed-effects logistic regression model. The two levels were individual and cluster(community) levels. It would be possible to use the flat logistic model by considering the effect of sample design using survey commands or also the cluster robust standard error methods, but it is not possible to see the variation between clusters using the latter methods. We ran four models to estimate both fixed effects of the individual and community-level factors and random intercept of between-cluster variation.

**Empty model:** This model was run without any factors, to test the random effect of between-cluster variability. Derived from the between-cluster and within-cluster variability, the intra-class correlation coefficient (ICC) was estimated to determine if the data justified using a multilevel approach for analyses by depicting the magnitude of between-cluster variability.

**Individual-level factors model:** The second model examined the effects of individual characteristics on pregnant women. Then ICC was estimated and observed if there was a decline in the between-cluster variability upon adding individual factors to the empty model.

**Community-level factors model:** This model contained only characteristics of clusters, not individuals. The unit of analysis for this model was the cluster.

**Combined model:** The important characteristics of individual women and clusters were concurrently fitted to one model to reveal their net fixed and random effects. The data were fitted into the model: this is explained below with short formula:

$$\log\left[\frac{\pi ij}{1 - \pi ij}\right] = \beta 0 + \beta 1 Xij + \beta 2 Zij + uj$$

Where;

✓ ij is the status of pregnancy (advanced age or not) in the $i^{th}$ pregnant woman at $j^{th}$ community

✓ $\pi ij$ is the probability of the presence of advanced-age pregnancy

✓ $(1-\pi_{ij})$ is the probability of non-advanced age pregnancy

✓ $\beta_0$ is the log odds of the intercept; it is the effect on the probability of advanced age pregnancy in the absence of predictors; and

✓ $\beta_1, \beta_2 \ldots$ and $\beta n$ are effect sizes of individual and community-level factors

✓ Xij. . . Z$_{ij}$ are independent variables at the individual and community level respectively

✓ uj showed the random effect (effect of the community on advanced age pregnancy for the
  *j*th community (cluster).

The random effect was explained using community variance, intra-class correlation ICC,
median odd ratio (MOR) and Proportional Change in Variance (PCV), whereas model fitness
was assessed by Log Likelihood (LL), Akaike's and Bayesian information criteria.

ICC was calculated using between-cluster variance and within-cluster variance
$\left\{ICC = Va + \pi\frac{2}{3}\right\}$. In log distribution, the residual variance of women within a cluster is
zero but the variance is considered constant at π2/3. The ICC was used to show the level of
cluster correlations within a model and to compare the successive models by looking at the
decline of the ICC value. In this study we use a STATA command "estat" after each model for
Intraclass correlation, information criteria and loglikelihood.

The Proportional Change in Variance (PCV) was computed for each model for the empty
model to show the power of the factors in the model to explain advanced-age pregnancy. it
was calculated as; $PCV = \frac{Va-Vb}{Va}$; Where Va is the variance in pregnant women in the empty
model and Vb is the variance in successive models.

The MOR is defined as the median value of the odds ratio between the area at the highest
risk and the area at the lowest risk of advanced-age pregnancy when randomly picking out two
areas and it depends directly on the area level variance. It can be calculated using the following
formula: $MOR = (exp\sqrt{2 \times 0.6745 \times v1}) \approx \exp(0.95\sqrt{v1})$. *V1* is the area-level variance, and
0.6745 is the 75[th] percentile of the cumulative distribution function of the normal distribution
with 0 mean and 1 variance. In this study, MOR shows the extent to which the individual prob-
ability of advanced-age pregnancy is determined by the residential area (primary sampling
unit (cluster).

### Ethics approval/clearance

Individual/household consent to participate in this study was "not applicable" in the study
since it is secondary data collected for EDHS, 2016. But, the permission to use the data for fur-
ther analysis was brought from http://www.dhsprogram.com. The data were analyzed and
reported in aggregate; household and individual identifiers were not reported.

## Results

### Individual socio-demographic characteristics of participants

As shown in Table 1 below nearly half of the respondents were Muslim followers 1,534
(46.58%), almost all were married by marital status 3,251(98.74%) and more than two-thirds
were non-educated 2,389(72.57%). The mean age at first sex and first marriage was 16.32±
0.051 and 16.66±0.065 years old respectively. Whereas the mean numbers of household mem-
bers, parities and under-five children were 6, 5 and 2 respectively. Nearly two-thirds of respon-
dents had no family planning unmet need 2,108(64.02%), and half of them had a poor wealth
index 1,645(49.96%).

### Community-level characteristics of participants

As shown in Table **2** below most of them were rural residents and from agrarian regions 3,074
(93.38%) and 2,946(89.47%) respectively. About 2,607 (80%) of the participants had higher
illiteracy and more than half of the participants had higher community poverty status 1,791
(54.41%).

**Table 1. Individual and household level characteristics of pregnant women in Ethiopian 2016 DHS (N[a] = _3,292_).**

| Individual level variables | Categories | Weighted | |
|---|---|---|---|
| | | Frequency | Percentage |
| Religion | Muslim | 1,534 | 46.58 |
| | Orthodox | 692 | 29.21 |
| | Protestant | 717 | 21.78 |
| | Catholic and other traditional | 80 | 2.44 |
| Marital status of | Currently in union | 3,251 | 98.74 |
| | Currently not in union | 41 | 1.26 |
| Age at first marriage (n = 3,290) | Mean and SD 16.66±0.065 | 95% CI:(16.54–16.79) | |
| Age at first sex (n = 3292) | Mean and SD 16.32±0.051 | 95% CI:(16.22–16.42) | |
| Mother's level of education | No education | 2,389 | 72.57 |
| | Primary | 785 | 23.83 |
| | Secondary and above | 188 | 3.60 |
| Partner's level of education | No education | 1,698 | 52.19 |
| | Primary | 1,224 | 37.61 |
| | Secondary and above | 332 | 10.20 |
| Women occupation | Have no work | 1,783 | 54.16 |
| | Have any types of work | 1,509 | 45.84 |
| Partner occupation | Agricultural works | 2,280 | 69.74 |
| | Employed | 733 | 22.41 |
| | Have no work | 257 | 7.86 |
| Listening Radio or watching TV | Not at all | 2,378 | 73.49 |
| | Less than once a week | 362 | 11.18 |
| | At least once a week | 497 | 15.35 |
| Individual level variables | Categories | Weighted | |
| | | Frequency | Percentage |
| Chewing chat | No | 2,660 | 80.78 |
| | Yes | 632 | 19.22 |
| Alcohol drinking | No | 2,562 | 77.83 |
| | Yes | 730 | 22.17 |
| Family planning unmet need | No unmet need | 2,108 | 64.02 |
| | Unmet for spacing | 507 | 15.41 |
| | Unmet for limiting | 677 | 20.56 |
| Total number of births | Mean and SD = 5.10±0.044 | 95% CI: (5.014–5.19) | |
| Age at first birth | 11–14 years old | 228 | 6.92 |
| | 15–19 years old | 2,011 | 61.08 |
| | 20–24 years old | 1,053 | 32.00 |
| Ever had a terminated pregnancy? | No | 2,925 | 88.84 |
| | Yes | 367 | 11.16 |
| Sex of household | Male | 2,912 | 88.46 |
| | Female | 380 | 11.54 |
| Relationship to Household | Head | 319 | 9.67 |
| | Wife | 2,838 | 82.20 |
| | Others* | 136 | 4.13 |
| Total household member | Mean and SD 6.28±0.038 | 95% CI:(6.21–6.36) | |
| Children under five at home | No | 443 | 13.47 |
| | One | 1,353 | 41.11 |
| | Two and above | 1,496 | 45.43 |

(_Continued_)

**Table 1.** (Continued)

| Individual level variables | Categories | Weighted | |
|---|---|---|---|
| | | Frequency | Percentage |
| Sons and daughters at home | No | 830 | 25.20 |
| | Yes | 2,463 | 74.80 |
| Sons and daughters elsewhere | No | 2,258 | 68.57 |
| | Yes | 1,034 | 31.43 |
| Sons and daughters died | No | 1,756 | 53.32 |
| | Yes | 1,536 | 46.68 |
| wealth index | Poor | 1,645 | 49.96 |
| | Middle | 644 | 19.55 |
| | Rich | 1,004 | 30.49 |

Others* = mother, sister, daughter, and others relatives and non-relatives a = Weighted samples

## Random effects and model fitness comparisons

As shown in Table 3 below the random effect was checked by intra class correlation (ICC), proportional change of variance (PCV), and median odd ratio (MOR). multilevel logistic regression model use in the analysis was justified by the significance of the community-level variance ($\sigma^2$ = 4.82 and p-value<0.001) that can be attributed to the cluster for being advanced age pregnancy, indicating there is significant differences between clusters on having advanced age pregnancy. The intra-cluster correlation coefficients also supported this, showing that 59.43% of the total variance of advanced age pregnancy in Ethiopia could be attributable to the context of the communities in which the woman belongs to. In the last model (model III), after adjusting for individual and community-level factors, about 48.67% of variations of advanced age pregnancy variation across communities were observed in the full model.

In the 4[th] Model after adjusting for individual and community-level factors Proportional change of variance revealed that 35.27%, which indicate that the variance in the odds of

**Table 2. Community-level characteristics of pregnant women from Ethiopian DHS 2016 in Ethiopia ($N^a$ = 3,292).**

| Community (cluster) level variables | Variable categories | Weighted | |
|---|---|---|---|
| | | Frequency | Percent |
| Place of residence | Urban | 218 | 6.62 |
| | Rural | 3,074 | 93.38 |
| Region | Agrarian regions | 2,946 | 89.47 |
| | Emerging regions | 305 | 9.26 |
| | Metropolitan cities | 42 | 1.27 |
| Community illiteracy | Lower | 685 | 20.82 |
| | Higher | 2,607 | 79.18 |
| Community poverty | Lower | 1,501 | 45.59 |
| | Higher | 1,791 | 54.41 |
| Community-women empowerment | Lower | 1,681 | 51.06 |
| | Higher | 1,611 | 48.94 |
| Community Media unexposed | Lower | 834 | 25.35 |
| | Higher | 2,458 | 74.65 |
| Community unmet need for family planning | Lower | 2,108 | 64.02 |
| | Higher | 1,184 | 35.98 |

Table 3. Random effects analysis for advanced-age pregnancy in Ethiopia based on Ethiopia demographic health survey 2016.

| Random effects | Null | Model I | Model II | Model III |
|---|---|---|---|---|
| Community variance | 4.82 | 4.03 | 4.28 | 3.12 |
| ICC % | 59.43 | 55.05 | 56.54 | 48.67 |
| PCV | Ref. | 16.39 | 11.20 | 35.27 |
| MOR | 2.09 | 1.91 | 1.97 | 1.67 |
| **Model fitness** | | | | |
| Akaike's Information Criteria | 2548.91 | 1314.82 | 2484.43 | 1458.73 |
| Bayesian Information Criteria | 2561.08 | 1460.80 | 2533.09 | 1276.26 |
| Log-likelihood (LL) | -1272.46 | -633.41 | -1234.22 | -608.13 |

advanced age pregnancy between communities was explained by individual and community-level factors found in the model. Moreover, Median Odd Ratio (MOR) for advanced age pregnancy was 2.09 in the null model, indicating heterogeneity between clusters. If we randomly select a mother from two different clusters, individuals at the cluster with a higher risk of advanced age pregnancy had 2.09 times higher odds of being advanced age pregnancy than individuals at the cluster with a lower risk of advanced age pregnancy. Model fitness was assured in the last model, where the lower values indicate the goodness of fit of the multilevel model.

## Determinants of advanced-age pregnancy

As shown in Table 4 below factors associated with advanced maternal age pregnancy in the last model; as maternal age increased at first birth were 4.05 times more likely had AAP (95% CI: 1.77–9.22), level of maternal education [primary education 2.72 times (95 CI: 1.55–4.76) and secondary education 5.65 times more likely had AAP compared with had no education (95% CI: 1.77–17.70)], having history of alcohol drinking 11.80 times more likely had AAP (95% CI: 5.71–24.42), as parity increased by one birth AAP were 3.22 times more likely (95% CI: 2.69 3.84), as number of household member increased by one AAP were 1.22 times more likely (95% CI: 1.05–1.41), unmet need for spacing of pregnancy had 4.79 times more likely had AAP compared with those have met need (95% CI: 2.63–8.74), having sons/daughters elsewhere 1.89 times more likely had AAP compared with those without (95% CI: 1.22–2.94), Living in metropolitan's city had 0.03 times less likely of AAP (95% CI: 0.03–0.38), those had higher community poverty and higher community unmet need for family planning had 2.37 times AAP (95% CI: 1.16–4.85), and 5.19 (95% CI: 2.72–9.92) times more likely of AAP, respectively.

## Discussion

This study was conducted to assess individual and community-level determinants of advanced maternal age pregnancy based on Ethiopian Demographic Health Survey data conducted in 2016. Even if there are a lot of studies on advanced maternal age and its adverse obstetrical and perinatal outcomes in both developing and developed countries, the factors that increase/decrease the trends of advanced maternal age pregnancy were not well studied including Ethiopia. Therefore, this study aims to identify individual and community-level factors that have a significant influence on the magnitude of advanced-age pregnancy in Ethiopia. The likelihood of advanced-age pregnancy was significantly higher among women who had a late age birth history (as the age of women increased by a year women were four times more likely to have AAP). Similarly maternal educational level [primary education 2.72 times and secondary

**Table 4. Multivariate multilevel logistic regression analysis for determinants of AAP from Ethiopian DHS 2016 in Ethiopia.**

| Individual/household level | Advanced age pregnancy (weighted sample size) | | | | | |
|---|---|---|---|---|---|---|
| | Yes | No | Null | Model I | Model II | Model III |
| Religion | | | | | | |
| Muslim | 447 | 1,086 | _____ | | _____ | |
| Orthodox | 371 | 590 | _____ | 0.33(0.16–1.67) | _____ | 0.19(0.09–1.42) |
| Protestant | 304 | 492 | _____ | 1.37(0.77–2.42) | _____ | 0.90(0.48–1.68) |
| Age at first birth (continuous) | | | _____ | 4.27(1.82–10.3)* | _____ | 4.05(1.77–9.22)** |
| Age at first sex (continuous) | | | _____ | 1.20(1.12–1.29)* | _____ | 1.22(1.13–1.31)** |
| woman education level | | | | | | |
| No education | 876 | 1,513 | _____ | | _____ | |
| Primary | 219 | 566 | _____ | 2.03(1.27–3.25)* | _____ | 2.72(1.55–4.76)** |
| 2ndary and above | 28 | 90 | _____ | 2.0(0.76–5.27) | _____ | 5.60(1.77–17.7)** |
| Women Occupation | | | | | | |
| Have no work | 568 | 1,215 | _____ | | _____ | |
| Have work | 555 | 954 | _____ | 0.84(0.57–1.23) | _____ | 0.93(0.62–1.40) |
| Drinking alcohol | | | | | | |
| No | 822 | 1,739 | _____ | | _____ | |
| Yes | 300 | 430 | _____ | 9.67(4.88–19.2)* | _____ | 11.8(5.71–24.42)** |
| Wealth index | | | | | | |
| Poor | 483 | 1,162 | _____ | | | |
| Middle | 253 | 391 | _____ | 3.25(1.98–5.34)* | | 1.59(0.85–2.97) |
| Rich | 387 | 617 | _____ | 2.99(1.78–5.03)* | | 1.42(0.72–2.79) |
| Parity | | | _____ | 2.80(2.39–3.29)* | | 3.22(2.69–3.84)** |
| Lists of household members | | | _____ | 1.30(1.13–1.48)* | | 1.22(1.05–1.41)** |
| Family planning unmet need | | | | | | |
| met need | 710 | 1,397 | _____ | | _____ | |
| Spacing Unmet | 271 | 236 | _____ | 1.64(1.02–2.63)* | _____ | 4.79(2.63–8.74)** |
| Limiting Unmet | 141 | 536 | _____ | 1.05(0.67–1.67)* | _____ | 1.64(0.95–2.84) |
| Ever had a pregnancy termination | | | | | | |
| No | 980 | 1,945 | _____ | | _____ | |
| Yes | 143 | 224 | _____ | 0.69(0.39–1.23)* | _____ | 0.66(0.36–1.22) |
| Sons/ Daughters at home | | | | | | |
| No | 124 | 705 | _____ | | _____ | |
| Yes | 999 | 1,464 | _____ | 1.47(0.87–2.49) | _____ | 1.15(0.65–2.02) |
| Son/daughter elsewhere | | | | | | |
| No | 528 | 1,729 | _____ | | _____ | |
| Yes | 594 | 440 | _____ | 1.99(1.31–3.02)* | _____ | 1.89(1.22–2.94)** |
| Number of under-fives at home | | | _____ | 0.27(0.21–0.35) | _____ | 0.26(0.20–0.35) |
| Advanced age pregnancy (weighted sample size) | | | | | | |
| Variables | Yes | No | Null | Model I | Model II | Model III |
| **Community level variables** | | | | | | |
| Region | | | | | | |
| Agrarians[a] | 1,039 | 1,907 | _____ | _____ | | |
| Emerging[b] | 79 | 226 | _____ | _____ | 0.81(0.26–2.51) | 0.29(0.14–1.59) |
| Metropolitans[c] | 5 | 37 | _____ | _____ | 0.2(0.028–1.38)* | 0.10(0.03–0.38)** |
| community illiteracy | | | | | | |
| Lower | 212 | 474 | _____ | _____ | | |
| Higher | 911 | 1,695 | _____ | _____ | 9.18(2.8–29.76)* | 1.88(0.89–3.98) |

(*Continued*)

**Table 4.** (Continued)

| Individual/household level | Advanced age pregnancy (weighted sample size) | | | | | |
|---|---|---|---|---|---|---|
| | Yes | No | Null | Model I | Model II | Model III |
| Community poverty | | | | | | |
| Lower | 601 | 900 | _____ | _____ | | |
| Higher | 522 | 1,269 | _____ | _____ | 6.3(2.13–18.39)* | 2.37(1.16–4.85)** |
| Media exposure | | | | | | |
| Lower | 784 | 1,520 | _____ | _____ | | |
| Higher | 339 | 650 | _____ | _____ | 0.90(0.30–2.68) | 0.70(0.33–1.50) |
| Community unmet need | | | | | | |
| Lower | 710 | 1,397 | _____ | _____ | | |
| Higher | 413 | 772 | _____ | _____ | 1.87(1.67–5.24)* | 5.19(2.72–9.92)** |

Notes

*significant at P<0.25 (crude); and

**significant at P<0.05 (adjusted)

[a]Tigray, Amhara, Oromia, and South nation, nationalities and people region

[b]Afar, Somali, Benishangul-Gumuz and Gambela

[c]Addis Ababa, Dire Dawa and Harari

education and above 5.65 times more likely to have AAP compared with had no education. This is supported by studies conducted in [45, 46]. The reason might be women who have a late first birth needs to have more children in their remaining reproductive age years and they spend more of their age on education. Besides this, they will also have late marriages as well. This leads most women to become pregnant at the age of 35 years and over.

In this study increase in the number of parity and household members, participants were 3.22 and 1.22 times more likely to have AAP, respectively. This is due to the cultural myths of the participants considering their child as wealth or as grace or blessing from God/Allah. They would have a pregnancy by the age of this. This study revealed that pregnant women who had a history of alcohol drinking have an 11.80 times higher chance of AAP. The possible explanation, after they have alcohol drinking, they might not control themselves or they decide poorly with the aid of alcohol.

Pregnant women who had a contraceptive unmet need for spacing of pregnancy had 4.79 times more likely AAP compared with those who have met the need. This is supported by [47]. Even an unmet need for spacing could not protect from advanced-age pregnancy, it would increase the number of women to become pregnant by this age. Finally, an important individual factor was had sons/daughters elsewhere were 1.89 times more likely to have AAP compared with those without. This might be due to feelings of loneliness and finding individuals that support them in their retirement age.

The odds of living in a metropolitan city had 0.03 times less likelihood of AAP. In such areas there might be higher community awareness about family planning utilization and higher socio-economic status leads women to good autonomy. There would be good women empowerment in decision-making than their counterparts. These findings also revealed that those with higher community poverty levels had 2.37 times more likely to Advanced Age Pregnancy compared to their counterparts. This might be due lack of infrastructure in the community to address sexual and reproductive health services, and the socio-economic status of the community surrounded by cultural constricts like considering children as wealth.

The other community factor determining Advanced Age Pregnancy in this study was those who had a higher community unmet need for family planning had 5.19 times more likely to

AAP among the community-level factors. This is also supported by a study done in Ethiopia [47]. This is due to the lack of modern family planning that is used for spacing pregnancy and/ or limiting pregnancy as a community.

## Strength and limitation

This is the first study to assess the determinants of advanced maternal age pregnancy among pregnant women in Ethiopia using Demographic and health survey data which is nationally representative data. The study used a multilevel logistic regression model which enhances the accuracy of estimates. Due to the cross-sectional nature of the data, a cause-effect relationship between outcome and independent variables could not be established.

## Conclusions

Maternal age at first birth, level of maternal education, history of alcohol drinking, parity, number of household members, family planning unmet need for spacing of pregnancy, had sons/daughters elsewhere, had higher community poverty level, those had higher community unmet need for family planning were more likely to have advanced age pregnancy. Whereas living in the emerging region and living in metropolitan's cities, were less likely to have advanced age pregnancy. Help women to have informed decision-making on the adverse effects that would happen during this age pregnancy. Create platforms for to women have special care during their preconception care, and antenatal care follow-up to minimize adverse effects on the mother and fetus/newborn. Address family planning unmet needs as an individual and community. Decrease community poverty by integrating different sectors like modernizing agriculture, tourism, investments, etc. breaking socio-cultural myths that women or community consider their children as wealth.

## Supporting information

**S1 File. Supplementary file for this study in STATA format.**
(DOC)

**S2 File.**
(DTA)

**S3 File.**
(DTA)

## Acknowledgments

We are grateful to DHS Measure, for providing the 2016 EDHS data for this study.

## Author Contributions

**Conceptualization:** Aynamaw Embiale Tesega, Degefa Gomora Tesfaye, Girma Geta, Muche Argaw.

**Data curation:** Aynamaw Embiale Tesega, Muche Argaw.

**Formal analysis:** Aynamaw Embiale Tesega, Degefa Gomora Tesfaye, Muche Argaw.

**Methodology:** Aynamaw Embiale Tesega, Aynadis Enyew, Degefa Gomora Tesfaye, Girma Geta, Alamirew Enyew Belay.

**Software:** Aynamaw Embiale Tesega.

**Supervision:** Muche Argaw, Alamirew Enyew Belay.

**Validation:** Aynamaw Embiale Tesega, Aynadis Enyew, Degefa Gomora Tesfaye, Girma Geta, Muche Argaw, Alamirew Enyew Belay.

**Visualization:** Aynamaw Embiale Tesega, Degefa Gomora Tesfaye, Girma Geta, Muche Argaw, Alamirew Enyew Belay.

**Writing – original draft:** Aynamaw Embiale Tesega, Degefa Gomora Tesfaye.

**Writing – review & editing:** Aynamaw Embiale Tesega, Aynadis Enyew, Degefa Gomora Tesfaye, Girma Geta, Muche Argaw, Alamirew Enyew Belay.

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
