## [Decision Letter · Decision Letter 0]

26 Feb 2024

PONE-D-23-41435Individual and community level Determinants of Advanced Age Pregnancy in Ethiopia, Multi-Level Analysis of National Data 2016PLOS ONE

Dear Dr. Tesega,

Thank you for submitting your manuscript to PLOS ONE. After careful consideration, we feel that it has merit but does not fully meet PLOS ONE’s publication criteria as it currently stands. Therefore, we invite you to submit a revised version of the manuscript that addresses the points raised during the review process.

We look forward to receiving your revised manuscript.

Kind regards,

Hamid Reza Baradaran, M.D., Ph.D.,

Academic Editor

PLOS ONE

Journal Requirements:

-https://doi.org/10.1186/s12884-020-2740-6

-https://doi.org/10.1089/jwh.2020.8860

In your revision ensure you cite all your sources (including your own works), and quote or rephrase any duplicated text outside the methods section. Further consideration is dependent on these concerns being addressed.

4. In the online submission form, you indicated that [data we used for this study will be Available on your request from the corresponding author.]

5. We note you have included a table to which you do not refer in the text of your manuscript. Please ensure that you refer to Table 1, 2 and 3 in your text; if accepted, production will need this reference to link the reader to the Table.

**Additional Editor Comments:**

There are some ambiguities in the employed model Please elaborate more details about the variables used in the statistical model

Reviewers' comments:

Reviewer's Responses to Questions

**Comments to the Author**

1. Is the manuscript technically sound, and do the data support the conclusions?

Reviewer #1: Partly

Reviewer #2: Partly

Reviewer #3: Yes

2. Has the statistical analysis been performed appropriately and rigorously? 

Reviewer #1: I Don't Know

Reviewer #2: Yes

Reviewer #3: No

3. Have the authors made all data underlying the findings in their manuscript fully available?

Reviewer #1: Yes

Reviewer #2: Yes

Reviewer #3: Yes

4. Is the manuscript presented in an intelligible fashion and written in standard English?

Reviewer #1: No

Reviewer #2: Yes

Reviewer #3: Yes

5. Review Comments to the Author

Reviewer #1: Method

In the data collection section, it is not clear in what period of time the data was collected. If the data is for childbirth in 2016, when were the other variables collected? Please clearly state the data collection date.

In the sample size measurement section, explain why 645 samples were taken and how many were selected in each cluster.

Explain more about the weighting and how it was.

In the eligibility criteria section, why were visitors who stayed at home the night before the survey included?

Results and discussion

In the results section, the tables must be given inside the text of the site. Also, below the tables, it should be specified what variables are included in the first model and how it differs from the third model.

Finally, it should be determined which model's efficiency and goodness of fit are better and which model's results should be used?

The results of Multilevel regression analysis and should be stated separately in a paragraph.

The manuscript needs language editing.

Reviewer #2: Title: It is better to express the title with a more measurable format.

Abstract: Please convey the objective of the study in a more measurable format. It is not necessary to show the analysis method in the objective of the study.

Keywords: OK

Introduction (background): there are a number of abbreviations which are not clear for what they are standing for, in contrast to their popularity (e.g.; AMA). I think it is good to express the reason you are using the DHS 2016 (seems to be an old one).

Material and method: Please note to the following issues in this section:

- Please define the proposed independent variables in more detail so that the measurement method will be clear.

- Please define how community level variables were collected and estimated. On the other hand, I could not understand why residence is considered as a community-level variable. I think it is also important to show how community-level variables were assigned to each individual. You have tried to explain the issue in the “Multilevel regression analysis” section, however, I think more definition is necessary. Maybe explaining the sampling method of the DHS especially the clusters, will suffice, especially in terms of the data and multilevel nature of it.

- There are a number of variables which are not defined in this section (e.g.; chewing chat or alcohol drinking). On the other hand, it is necessary to show the reliability of the responses as well.

- Please show the software used for the analysis of the data, together with the statistical significance level of the study.

Results: It is necessary to show the proposed characteristics in each group of advanced age pregnancy and the other one separately. Showing the estimated effect measure is not enough with this regard.

Discussion: OK; however, a number of revisions may be required following revising of the previous sections.

References: OK

Reviewer #3: The study in Ethiopia examined factors contributing to advanced maternal age pregnancies utilizing data from the 2016 Ethiopian Demographic and Health Survey. It involved analyzing 3,292 weighted samples of pregnant women. Using a multilevel logistic regression model, the research identified that increased maternal age at first birth, lower levels of maternal education, history of alcohol consumption, greater parity (number of children), unmet needs for family planning, having children residing outside the home, higher community poverty levels, and unmet community needs for family planning were associated with an increased likelihood of advanced-age pregnancies. Conversely, being located in an emerging region or living in urban centers like metropolitan cities showed negative associations with advanced-age pregnancies. However, there are several questions in this study.

1. In statistical analysis, "bivariate" usually refers to examining the relationship between two variables at the same time. I did not understand what the authors meant in this study.

2. The statement "mothers are nested within households, and households are nested within clusters" suggests a hierarchical structure where individuals (mothers) are grouped within households, which in turn are grouped within larger clusters or communities. While a three-level model could have been more appropriate to account for this nested structure, the authors opted for a two-level model in this study. This decision may have overlooked the potential influence of the family unit on advanced maternal age pregnancies.

3. Interview date can indeed be considered a confounder in the models if it is associated with both the exposure (factors contributing to advanced maternal age pregnancies) and the outcome (advanced-age pregnancies). If interview date is related to any of the variables being studied and could potentially influence the results, it should be accounted for in the analysis to ensure accurate and unbiased findings.

5. In the Combined model:

a. The formula does not specify the meaning of variables i and j.

b. Variables i and j are not clearly labeled under the index.

c. Uncertainty exists regarding whether β_0 represents the width from the overall intercept.

d. suggest the authors to use distinct variables for individual and social independent variables such as X and w.

e. The presence of random errors (u_j) raises questions about the location of random effects within the model.

f. The distribution of u_j remains unknown, adding to the uncertainty surrounding the model's components.

6. The authors' use of "log distribution" was unclear. It should refer to a standard logistic distribution with variance.

7. The statement regarding the residual variance of women within a cluster in log distribution should be clarified to involve a power term, not just a constant.

8. Please provide the PCV formula.

9. It is suggested to calculate MOR (Mean Odds Ratio) and add it to Table 3.

10. Please add the ICCcom formula and guidance on its interpretation to decide on the analysis method.

11. STATA software lacks multilevel models, prompting a discussion on the statistical analysis methods or software packages utilized in the study.

12. Models referenced in Table 3 require further elaboration and description.

6. PLOS authors have the option to publish the peer review history of their article (what does this mean?). If published, this will include your full peer review and any attached files.

Reviewer #1: No

Reviewer #2: **Yes: **Babak Eshrati

Reviewer #3: **Yes: **Safdar Masoumi

---

## [Author Response · Author response to Decision Letter 0]

10 Apr 2024

To the editors; we are paraphrasing and rewriting for language issues. tables inserted in the text with cross-reference format in the text body. variables also clearly elaborated in the variable section of the manuscript. all PLOS formats also applied on headings, sub-headings depend on the PLOS protocol. 

To reviewer; we were used all of our efforts to responded to your genuine and constructive comments and questions. thank you for your time and genuine concerns. we are attaching the responses titled "responses to reviewer."

---

## [Editor Report · Decision Letter 1]

22 May 2024

DETERMINANTS OF ADVANCED AGE PREGNANCY IN ETHIOPIAN; MULTI-LEVEL ANALYSIS OF ETHIOPIAN DEMOGRAPHIC HEALTH SURVEY 2016

PONE-D-23-41435R1

Dear Dr. Tesega,

We’re pleased to inform you that your manuscript has been judged scientifically suitable for publication and will be formally accepted for publication once it meets all outstanding technical requirements.

Kind regards,

Hamid Reza Baradaran, M.D., Ph.D.,

Academic Editor

PLOS ONE
---

## [Editor Report · Acceptance letter]

14 Jun 2024

PONE-D-23-41435R1 

PLOS ONE

Dear Dr. Tesega, 

I'm pleased to inform you that your manuscript has been deemed suitable for publication in PLOS ONE. Congratulations! Your manuscript is now being handed over to our production team.

Kind regards, 

on behalf of

Professor Hamid Reza Baradaran 

Academic Editor

PLOS ONE